# Multiple Post-SARS-COV2 Infectious Complications in Kidney Transplant Recipient

**DOI:** 10.3390/medicina58101370

**Published:** 2022-09-29

**Authors:** Patrycja Grzejszczak, Agnieszka Płuciennik, Anna Kumor-Kisielewska, Ilona Kurnatowska

**Affiliations:** 1Department of Internal Medicine and Transplant Nephrology, Medical University of Lodz, 90-153 Lodz, Poland; 2Department of Nephrology, Barlicki Memorial Teaching Hospital No. 1, 90-153 Lodz, Poland; 3Department of Pathobiology of Respiratory Diseases, Medical University of Lodz, 90-153 Lodz, Poland

**Keywords:** kidney transplantation, SARS-CoV-2, multi-pathogenic infections

## Abstract

A forty-seven-year-old recipient in late period after kidney transplantation with chronic estimated glomerular filtration rate (eGFR) 30 mL/min/1.73 m^2^, fully vaccinated against COVID-19 was diagnosed with SARS-CoV-2 infection in November 2021. After an initially mild course of the disease, he developed multiorgan failure requiring periodic respiratory and dialysis therapy. Covid-19 disease was complicated by multiple infections such *Clostridioides difficile* infection, *Streptococcus epidermidis* bacteriemia, *Klebsiella pneumoniae* and *Candida glabrata* urinary tract disease, cytomegalovirus infection and oral candidiasis. In a short period, he was readmitted to the hospital twice with recurrent *Klebsiella pneumoniae* urosepsis. One of those hospitalizations was also complicated by another COVID-19 infection that was confirmed with non-reactive neutralizing antibody. Due to severe infections the patient required individualized modification of immunotherapy; however, due to their recurrence it was finally decided to be discontinued. The patient was also reintroduced to hemodialysis therapy and no infections occurred since then.

## 1. Introduction

Infections, besides cardiovascular complications and cancers, are the most common cause of morbidity and mortality in patients after kidney transplantation (KTx) [1]. The immunosuppressive therapy is the main risk factor of development and severe course of infections. Other risk factors include, e.g., impaired graft function, malnutrition, immunosuppressive therapy, and long history of dialysis before KTx, etiology of kidney failure, diabetes mellitus, chronic colonization of bacteria or viruses, presence of chronic catheters [2,3]. The etiology of infections may vary and depends on the length of period after KTx. Moreover, presence of one pathogen may promote infections by other agents. In the early period after transplantation infections that occur are typical such as wound infection, pneumonia or urinary tract infection [UTI]. Between the first and six months after KTx, the usual infections caused by opportunist microorganisms or latent viruses reactivation are diagnosed. In a later period, during the maintenance immunosuppression therapy, infections typical of general population are most common such as UTI or pneumonia with a community usual pathogens etiology [3]. Nowadays a new virus SARS-CoV-2 has also become a problem in KTx. The prevalence of COVID-19 infection in KTx population is unknown; however, this group is at increased risk of developing severe acute respiratory distress syndrome. Many reports show that KTx recipients hospitalized with SARS-CoV-2 have high mortality rates approaching 20–32% [4,5,6] and reaching 45% in those admitted to intensive care units (ICU) [7]. It should be emphasized that lower efficacy of vaccination against COVID-19 in this population results from the immunosuppressive treatment and is also a risk factor of severe course of COVID-19 [8,9]. Those data suggest that we should look not only for appropriate COVID-19 treatment but also ways to properly modify immunosuppression therapy during active infection and perhaps vaccinating time, which in KTx still remains unclear [4,10]. 

We present a case of a patient in long term after KTx with severe course of SARS-CoV-2 infection complicated by multi-pathogenic infections. 

## 2. Case Report

The patient is a 47 year old man, 11 years after KTx from deceased donor with suboptimal graft function, estimated glomerular filtration rate (eGFR) chronically 25–35 mL/min/1.73 m^2^), with artificial mitral valve because of endocarditis in 2004 (treated with vitamin K antagonists) and with history of arterial hypertension. His maintained immunosuppression regimen consisted of tacrolimus (TAC; with trough levels 5–8 ng/mL), mycophenolate mofetil (MMF; daily dose 1000 mg), and prednisone (5 mg a day). He was vaccinated against COVID-19 with two doses of Pfizer (on April and May 2021) and third dose of Moderna (September 2021). Nevertheless, in November 2021 he was diagnosed with COVID-19 infection (PCR test was positive) with initially mild symptoms (weakness, cough, temperature to 37.8 °C). He was treated symptomatically at home, but after 2 weeks a significant deterioration of general condition occurred and he was admitted to the covid unit with cough, hemoptysis, dyspnea, fever up to 40 °C, chills, and decreasing urination. Due to hemodynamic instability and multiorgan failure: tachycardia 130–140/min; hypotension (60/49 mmHg), and saturation 77% (with no oxygen therapy) he was admitted to ICU. His laboratory tests revealed respiratory failure, serious anemia, low platelets, elevated C-reactive protein (CRP) and pro-calcitonin (PCT) levels with high IL-6 level (432 pg/mL; N: <7), deep coagulation disturbances, and significant deterioration of graft function (see Table 1 part A and B).

His chest computer tomography (CT) showed wide areas of bilateral ground-glass opacities with segmental areas of crazy paving and local inflammation (Figure 1). Respiratory therapy and continuous hemodialysis (HD) were implemented and he received dexamethasone in dose of 12 mg per day with wide-spectrum antibiotic therapy (meropenem and vancomycin, followed by linezolid), MMF was stopped at once while TAC dose was first reduced and then discontinued a few days later. He required red blood cells (RBCs) transfusion. Remdesivir was not administered due to a long interval between diagnosis of COVID-19 infection and treatment, nor was tocilizumab due to high PCT levels. After two negative Abbott COVID-19 tests, his isolation was ended. In the third week of hospitalization an improvement of general status was observed, patient was extubated, his diuresis gradually resumed and graft function improved so HD was stopped and he was transferred to nephrology department two days later for further treatment in general semi-severe condition. On admission to nephrology, he was cachectic, lying down with visible dyspnea at any movement and still dependent on oxygen mask with reservoir, with oxygen flow 14 L/min. There were also present symptoms of oral candidiasis and diarrhea of clostridium difficile etiology so the treatment with itraconazole and vancomycin orally and metronidazole intravenously was started. On admission his inflammatory markers were high (CRP 353.6 mg/L; PCT 4.13 ng/mL; see Table 1. Part C) so he was tested and found positive for both bacterial (*Enterococcus faecium*, *Klebsiella pneumoniae*) and fungal (*Candida glabrata*) infection in the urinary tract as well as blood infection (*Staphylococcus epidermidis*). Targeted antibiotics (teicoplanin, meropenem) were administered. CMV DNA was also detected and valganciclovir was started. Throughout the whole hospitalization, the patient required rehabilitation and nutritional treatment with gradually expanded oral diet. The therapy resulted in improvement of his general status with disappearance of inflammation signs and dyspnea. During hospitalization RT-PCR COVID-19 tests were performed four times as per local standards—all were negative; IgM and IgG concentrations were not assessed. The patient had diuresis (2 L/day) with satisfactory stable graft function (serum creatinine [sCr] 2.5–3.2 mg/dL) and immunosuppressive treatment was implemented again, first TAC with trough level around 5 ng/mL and dexamethasone was converted to prednisone. Then MMF treatment was restarted in reduced dose (500 mg/day) on discharge on day 39 of hospitalization.

However, 7 days later he was readmitted to nephrology department with signs of septic shock. There were clinical presentations of fever, hypotension, and confusion with anuria. Laboratory tests (Table 1 part D) once again revealed progression of graft failure with deep metabolic acidosis together with high CRP and PCT levels, anemia and coagulation disturbances. Abdomen ultrasonography revealed graft edema with suspicion of graft abscess (excluded in next days). Endocarditis was excluded by echocardiography. MMF as well as VKA were stopped (low molecular weight heparin (LMWH) was started instead), meropenem with teicoplanin was instituted empirically and meropenem was maintained according to antibiotic sensitivity (both in blood and urine cultures K. pneumoniae were isolated). The patient required RBCs transfusion again. Moreover, due to graft failure with deep metabolic disturbances and urosepsis daily HD therapy was implemented. With the treatment the general condition improved, however graft function did not return to the previous level remaining around 4.9 mg/dL of serum creatinine (eGFR 14 mL/min/1.73 m^2^) with 2.5 L/d diuresis so HD was suspended. Apart from that all control laboratory tests results were improving (Table 1 part D). Due to COVID-19 infection in the ward the patient had a swab rRT-PCR test and was again positive. He also had positive antibodies against SARS-CoV-2 both IgM > 30.0 AU/mL (negative-non-reactive: <3.5 AU/mL) and IgG 3.14 AU/mL (non-reactive: ≤1.1 AU/mL), but no symptoms of COVID-19 infection. Since the patient had been vaccinated against SARS-CoV-2, had undergone COVID-19 infection two months earlier, and the antibodies pattern showed a slight increase in IgG titer, the neutralizing antibody titer was assessed and found to be non-reactive with antibody concentration 0.247 g/mL (reactive result ≥ 300 g/mL), a new COVID-19 infection was diagnosed, but due to its asymptomatic course no antivirus treatment was instituted. After isolation for 10 days, the patient was discharged with immunosuppressive treatment only with TAC and prednisone 15 mg/day (without MMF). Once more, due to recurrence of UTI and sepsis symptoms he was readmitted 2 weeks later with progressing graft failure and septic laboratory markers (Table 1 part E). *K.pneumoniae* urosepsis was confirmed again by blood and urine cultures and meropenem and fluconazole as a fungal reinfection prophylaxis were resumed. Due to recurrence of severe infections in the patient with a significantly impaired graft function and with artificial mitral valve it was decided to discontinue immunosuppressive treatment and implement chronic HD. After two weeks, the patient was discharged in good general state and no more infectious complications have been diagnosed since then. After six months of observation, he is planned to be vaccinated with a fourth dose against COVID-19 and reported to the national kidney transplant list.

## 3. Discussion

We present a case of a multi-pathogenic and recurring infection in a KTx patient which manifested as a series of events that followed COVID-19 infection. Our patient presented typical risk factors for severe course of COVID-19 disease observed both in general population [11] and in solid organ transplant recipients [10,12,13], such as male sex, history of hypertension, and immunosuppression treatment. Moreover, worsening graft function is considered to be a risk factor as well [7]. One of the reasons we present the case is that the patient developed SARS-CoV-2 severe infection despite the full program of vaccination, second he was diagnosed with COVID-19 infection twice in a quite short period (two months) and those infections and their treatment were complicated by multi-pathogenic co-infections: multibacterial, fungal, and CMV. He also three times developed septic shock. Only the withdrawal of immunosuppressive treatment halted the development and recurrence of infections. 

Clinical presentation of SARS-CoV-2 in KTx recipients ranges from mild to severe, including multiorgan failure and death [12]. Our patient presented typical and most common symptoms of COVID-19 disease observed both in general [11] as well as in KTx population [10,14,15] before finally developing acute respiratory distress syndrome (ARDS) which is one of the most critical manifestations [14]. Maybe in order to prevent further complications we should have recommended at that stage a modification of immunosuppressive treatment (MMF dose reduction together with the increasing dose of glucocorticosteroids) and conversion of VKA to LMWH. However, due to a remote place of residence, the patient was not able to come to the outpatient posttransplant clinic so the laboratory tests and full medical investigation could not be performed at the specialist center earlier and only symptomatic treatment was recommended. Lymphopenia, which was observed in our case, is common in persons infected with COVID-19 in general population [11] as well as in KTx [15] and is associated with severity of the disease [16], but it can be a consequence of immunosuppressive therapy as well [17]. Our patient had decreased platelet count and elevated CRP, PCT, D-dimer, lactate dehydrogenase (LDH), creatine kinase (CK), and creatinine which in meta-analysis of 32 studies including 10,491 patients with confirmed COVID-19 are considered to be indicators of poor outcomes, as well as elevated ALT and AST [18]; however, the latter two parameters were normal in the presented case. Bilateral infiltrations with the image of frosted glass in our patient’s lungs are most common abnormalities described in patients with SARS-CoV-2 infection [14]. The most frequently observed complication of COVID-19 in general population is sepsis caused by bacteria, followed by ARDS and multiorgan insufficiency including kidney failure [11] which all occurred in herein presented patient. At the beginning of COVID-19 infection our patient, due to mild symptoms, was treated symptomatically at home and no modification of immunosuppression was recommended, but as his general condition worsened and he was eventually admitted to ICU, MMF was suspended together with reduction of the TAC dose and discontinued when the patient required respiratory therapy. Such procedure is recommended by experts although different centers have different strategies with regard to the risk of graft rejection [9]. That is why it is necessary to formulate proper management of immunosuppression and further guidelines in the treatment of COVID-19 in the transplant population. Our patient did not receive remdesivir because of a long interval between diagnosis and treatment of COVID-19, nor tocilizumab due to high PCT levels. He was treated only with dexamethasone and received broad-spectrum antibiotic therapy. 

The deep coagulation disturbances might have been caused by septic shock, graft failure or interaction between acenocumarol and TAC (both are metabolized by cytochrome P450). 

Our patient developed severe COVID-19 disease despite taking three doses of vaccination and was diagnosed with COVID-19 for a second time after experiencing SARS-CoV-2 infection although without symptoms. His exact serological status before SARS-CoV-2 was unknown but immunocompromised patients, including KTx recipients usually have worse response to vaccination [8,9]. However, a study by Tylicki et al. shows that the third dose of vaccine in KTx population produces a large increase in the number of antibodies, achieving a titer almost 30% higher than after the second dose [8]. Unfortunately, we do not know our patient’s serological status against SARS-CoV-2 (it was not checked before infection) but we do know that he was fully vaccinated and, what is worth emphasizing, with a more effective combination of vaccines (twice with Pfizer and once with Moderna). Unfortunately, there is a lack of recommendations to assess the serological answer to vaccination against SARS-CoV-2 infection in immunocompromised patients. Currently, the monoclonal antibodies (MAB) against COVID-19 are available as a pre-exposure prophylactic medication in immunocompromised patients and it may be an option in preventing a serious course of disease in solid organ recipients. The MAB targeting the SARS-CoV-2 spike protein may be an option in the COVID-19 infection treatment in Ktx patients, which has specifically been evaluated as an early outpatient treatment to prevent the clinical progression of mild to moderate COVID-19. Studies suggest that early intervention with MAB treatment may reduce the necessity of emergency department visits or hospitalization for COVID-19, especially in high-risk patients [19]. This treatment seems to be safe both for patients and for graft. However, this kind of treatment was not available at the moment of our patient’s infection. From July 2022 in Poland the monoclonal antibodies: tiksagewimab and cilgawimab are available and recommended as pre-exposure prophylactic medications. These antibodies can provide protection for many who cannot achieve a sufficient immune response to the vaccine, including solid organ recipients [20]. Perhaps administration of MAB should be considered in recipients unresponsive to vaccination against COVID-19. 

Furthermore, in less than two months after the first infection, during a routine check-up of patients hospitalized in Nephrology Department our patient had a positive result from the swab PCR test. Due to the lack of specific symptoms of SARS-CoV-2 infection and the possibility that the residual genes from previous coronavirus infection remained in the blood circulation (PCR tests in immunocompromised patients may remain positive up to several months), we identified antibodies as well as neutralizing antibodies that both confirmed the new infection. That result clearly indicated lack of neutralizing effect despite the previously acquired immunization. It should be noted that the SARS-CoV-2 neutralizing antibody prevents invasion of human cells by SARS-CoV-2 by inhibiting or even neutralizing the biochemical effects of the virus. The SARS-CoV-2 neutralizing antibody reduces the infectivity of the virus by binding to surface epitopes of virus particles and blocking virus entry into the infected cell. Detection of SARS-CoV-2 neutralizing antibodies may reflect the body’s immune response to SARS-CoV-2 and ultimately determine whether there is antibody protection or immunity. Some SARS-CoV-2 binding antibodies may not have neutralizing activity. Therefore, detection of SARS-CoV-2 neutralizing antibodies is an important way to assess the immune status of SARS-CoV-2. It should be noted that both the type of vaccine used and the mutation of the virus do not affect the results of the test used [21,22]. The results of the tests confirmed that our recipient had two SARS-CoV-2 infections one after the other. We did not have an opportunity to assess the genotype of the virus but the Omicron variant was suspected, especially that the course of the infection was mild and Omicron was a more frequent cause of infection in that period.

COVID-19 infection and probably its treatment and long hospitalizations were the cause of multi-pathogenic co-infections in the immunocompromised individual. The main risk factors of developing such complications aside from immunosuppressive treatment are ICU hospitalization with intubation and graft failure with a later HD treatment, high doses and long treatment with dexamethasone and long wide-spectrum antibiotics therapy, presence of central and Foley catheter, blood transfusions, cachexia, and general long hospitalization. Some studies show that among solid organ transplant recipients with COVID-19 infection, bacterial co-infections are most common (50%), followed by CMV (37,5%) and fungal infections (8.3%) [23]. Our patient was diagnosed with septic shock a few times in a short period of time. It is worth mentioning that in an immunocompromised person, septic shock may develop really quickly and much faster than in a non-immunocompromised patient, that is why careful observation, proper diagnostics, and treatment are necessary. 

It is difficult to differentiate whether all those infections were complications of hospitalization itself or the state of immunosuppression together with severe course of COVID-19 illness followed by multiorgan dysfunction and development of malnutrition despite nutritional treatment. Nevertheless, despite presenting nearly all epidemiological and laboratory factors of poor prognosis followed by multiple infectious complications the patient survived. Decreasing immunosuppression treatment should be the first step in infection therapy although it may increase the risk of graft rejection. In the presented case, after taking into account the overall clinical picture of recurrent severe multi-pathogenic infections in the patient treated with immunosuppressive therapy with significantly impaired graft function and with an implanted artificial heart valve, it was decided to discontinue the immunosuppressive treatment and reintroduce HD therapy. The applied treatment improved the patient’s general condition. Now the patient has been HD for ten months, he is in good clinical condition, no signs of infection have been observed, and he is prepared for a second kidney transplantation. 

## 4. Conclusions

Patients after kidney transplantation are at high risk of developing severe infections, and the presence of one pathogen paves the way for other infections, therefore the patient should be carefully monitored and the coexistence of other infections should be taken into account, especially in patients with COVID-19 infection. In KTx patients with severe course of infections, the reduction of immunosuppression or even discontinuation in patients with impaired graft function and return to hemodialysis should be considered. Consequently, such may not only save the patient’s life but also give the prospect of another kidney transplant in the future.

## Figures and Tables

**Figure 1 medicina-58-01370-f001:**
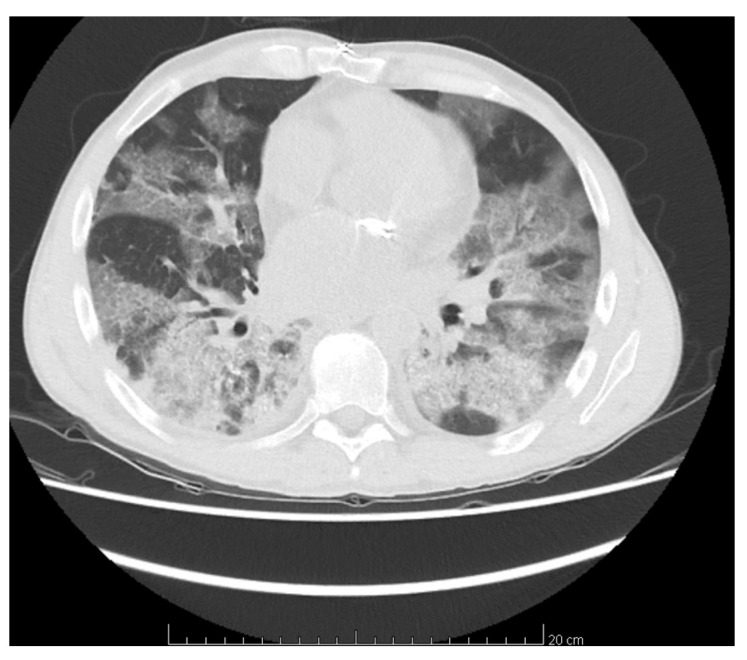
Computer tomography scan revealed wide areas of bilateral ground-glass opacities with segmental areas of crazy paving and local inflammation.

**Table 1 medicina-58-01370-t001:** The laboratory results during hospitalizations in: covid unit (A), intensive care unit (B), and nephrology department (C, D, and E).

	Covid Unit (A)	Intensive Care Unit (B)	Nephrology Department 1st Hospitalization (C)		Nephrology Department 2nd Hospitalization (D)		Nephrology Department 3rd Hospitalization (E)	
Parameter/Time period			I	II	I	II	I	II
HGB [g/dL]	8.9	5.0	8.9	10.6	9.1	8.9	9.2	7.2
RBC [T/L]	2.83	1.69	2.94	3.57	3.11	3.08	3.17	2.45
HCT [%]	26.2	14.9	27.3	32	27.9	26.9	29.6	23.1
WBC [G/L] Lymphocyte [%] Neutrophile [%]	7.7 6.2 87.4	7.7 5.9 88	10.9 4.9 89.5	7.4	18.7 3.8 91.7	6.9	15	8.7
PLT [G/L]	106	82	110	146	229	124	192	180
CRP [mg/L] [N: <5.0]	187	144	353	26	255	36	230	19
Procalcitonin [ng/mL] [N: 3.5–5.1]	5.74	22.37	4.13		34.4		14.3	
eGFR [mL/min/1.73 m^2^]	10.5	10.0	>60 (after SLEDD)	22	7	14	8	13
Creatinine [mg/dL] [N: 0.72–1.18]	5.9	6.2	1.13 (after SLEDD)	3.26	8.9	4.9	7.4	4.5
Urea [mg/dL] [N: 10–50]	229	315	72 (after SLEDD)	108	268	155	176	116
Albumin [g/L] [N: 35–52]	27	18.2	21.4	22.9				
Total Protein [g/L] [N: 60–80]		36.8	55.9	50.4				
LDH [U/L] [N: <248]	940	853	664					
D dimer [ng/mL] [N: <500]	242	1507	1438		1070			
INR	No reaction	1.24	1.08	3.24	7.25	1.1	8.92	1.22
APTT [s]	No blood clotting	45.7	44.1	50.9	141.1	58.6		42.4

APTT—activated partial thromboplastin time, CRP—C-reactive protein, eGFR—estimated glomerular filtration rate, HCT—hematocrit, Hgb—hemoglobin, LDH—lactate dehydrogenase, PLT—platelets, RBC—red blood cells, SLEDD—sustained low-efficiency daily dialysis, WBC—white blood cells. Period I shows laboratory tests on admission, period II on discharge.

## Data Availability

Data sharing is not applicable.

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
