# Peer review of "Multiple Post-SARS-COV2 Infectious Complications in Kidney Transplant Recipient"

_medicina, 2022, doi:10.3390/medicina58101370_

Round 1

Reviewer 1 Report

A well-written case report about recurring co-infections in a KTx recipient following Covid-19 infection dealt with by discontinuing IS. The case is well presented and the conclusions mostly justified by the general literature and the specifics of the case presented. The main weakness is that it is, in fact, just a case report, and conclusions a tough to defend based on one complicated case. I do however feel that the quality of the presentation is high and that it does add something to the body of literature already published on Covid-19 about the KTx cohort. Another weakness admitted by the authors is the lack of pre-infection serology. I would like the authors to focus the manuscript slightly more on what could have been done to avoid the initial deterioration and thus the subsequent spiral of co-infections and organ failure. While the management of the patient resulting in cessation of IS seems sound - an ounce of prevention would have been worth a pound of cure. What thoughts do the authors have about that?

If available I would like the authors to report the cycle threshold value of the first PCR test to elucidate the initial viral load. Also, comment on what follow-up was done on this patient in the outpatient setting. He apparently only was admitted after becoming quite sick with a co-infection. Maybe comment in the discussion on options such as the importance of intensive outpatient follow-up of transplanted patients with Covid-19 infections to avoid transitioning from outpatients to inpatients with severe disease. Also, comment on the use of monoclonal antibody therapies as an option for transplanted patients to avoid a severe course. I would also like to know about the definitive or probable genotype of the virus affecting the patient. November 2021 was just before the transitional period between delta and omikron.  Can we assume both Covid-19 infections were delta? Please comment on local epidemiology to clarify this point. 

Author Response

Answers to reviewer

Thank you very much for your comments.

As for what could have been done to avoid the initial deterioration and in order to prevent further complications we should have recommended at that stage a modification of immunosuppressive treatment (MMF dose reduction together with the increasing dose of glucocorticosteroids) and conversion of VKA to LMWH. However due to a remote place of residence, the patient was not able to come to the outpatient posttransplant clinic so the laboratory tests and full medical investigation could not be performed at the specialist center earlier and only symptomatic treatment was recommended by phone call.

Unfortunately,  there is lack of recommendations to assess the serological answer to vaccination against SARS-CoV-2 infection in immunocompromised patients. Currently, the monoclonal antibodies (MAB) against COVID-19 are available as a pre-exposure prophylactic medication in immunocompromised patients and it may be an option in preventing a serious course of disease in solid organ recipients.  We have also added the discussion information about using MAB in organs recipients.

During hospitalization and follow-up we discontinued the immunosuppressive treatment (MMF was stopped, GCS gradually discontinued and TAC dose reduced), HD therapy was maintained and at the moment patient is being prepared for second kidney transplantation.

Unfortunately we do not know the cycle threshold value of the first PCR test and also in Poland at that time we did not have an opportunity to assess the genotype of the virus but the Omicron variant was suspected, especially that the course of the infection was mild and Omicron was a more and more frequent cause of infection in that period  (we have added this information to the discussion).

Reviewer 2 Report

Grzejszczak et al, presented a case of fulminant infection after covid in a renal transplant recipient. The case is well written, however the significance is low. There is lack in evidence that support the need to stop immunosuppression to prevent recurrent COVID infection. There is also lack of evidence that multi-infection and recurrent urosepsis in not due to hospital acquired vs community acquired infection. Furthermore the need for triple immunosuppression in late post transplant period is questionable. 
The baseline GFR of the patient was poor and probably which might raise question that the patient was receiving un-necessary immunesuppression. The article did not show whether the patient was evaluated for chronic rejection or not. 

Author Response

Answers to reviewer

Thank you very much for your comments.

Indeed there is lack of evidence that modification or stopping immunosuppression prevents COVID-19 recurrence. There are some data that minimization of immunosuppression, mainly reduction or MMF withdrawal in moderately severe course of the disease, together with reduction or withdrawal of TAC with dexamethasone implementation in heavy course of infection should be considered.

In our opinion the multiple infections in our patient may have been caused by immunodeficiency (CKD, immunosuppressive drugs including large doses of GCS) and the fact that one infection promoted development of another or  such complications were just  hospital acquired (procedures, catheters etc) and were connected with long hospitalization and all medical procedures and concomitant treatment in our case.

Patient was treated with triple immunosuppression in late post-transplant period, however, from the beginning after transplantation his graft function was impaired but with stable levels of serum creatinine (2-2.5 mg/dl) and eGFR oscillating 25-35 ml/min/1.73m2. The patient was a young person, he also required high doses of TAC to maintain the correct TAC level (he was a fast TAC metabolizer). There are no recommendations for discontinuation of e.g. MMF immunosuppressive treatment in such patients, although our patient received adequately small doses (with TAC levels about 5 ng/ml, MMF 1g/day and prednisone 5mg/day). Until COVID-19 infection his graft function (for 11 years) was completely stable and there were no data or symptoms for a rejection so he was not diagnosed for it.

Round 2

Reviewer 2 Report

Thank you for your response and having the time to modify your manuscript according. I have few comments to add. 

Comment 1: 

The case-study has multiple limitations. I would like to thank the authors for acknowledging the limitation of the study. 

Comment 2: 

Suggestion of return to dialysis in the conclusion, might give a false impression that fulminant infection is a cause of graft loss. Furthermore no data for long-term follow-up or consideration of graft loss and preparation for re-transplantation. 

Comment 3: 

I would like to thank the authors for acknowledging the study limitation. However, the manuscript presents multiple areas that there is suggestions of what should have done, and whether that would have changed the management course. The latter decreases the scientific soundness of the article. 

Comment 3: 

The patient is CKD by definition having a GFR of 25-35 ml/min/1.73m2. This patient would any way go to dialysis at a point during the coming period. The case doesn't mention whether there was a trend down for the GFR after transplantation, and was there any  drastic change of the GFR drop was subtle. 

Author Response

Answer to Reviewer

Comment 1: 

The case-study has multiple limitations. I would like to thank the authors for acknowledging the limitation of the study. 

Thank you for your comments. We know that the case-report has a lot limitations. We decided to publish the description to draw the reader's attention to the need to carefully monitor a transplant recipient  with one infection for co-infections and because the presence of one pathogen paves the way for other infections. We would like to show that the main risk factor of infection recurrence was the immunosuppression treatment - possibly excessive, perhaps it should have been discontinued earlier in our patient, but the return of graft function after stopping continuous dialysis in IUC, and the patient's reluctance to return to dialysis treatment was the cause of decision to return to immunosuppression after discharge from IUC.

Comment 2: 

Suggestion of return to dialysis in the conclusion, might give a false impression that fulminant infection is a cause of graft loss. Furthermore no data for long-term follow-up or consideration of graft loss and preparation for re-transplantation. 

In our opinion the multiple infections, their complications and treatment was a cause of graft loss in our patient. His graft function was impaired but stable during many years after KTx (eGFR chronically 25-35 ml/min/1.73m2) till the described Covid infection and their complications. The data about patient follow-up are included in the manuscript, but maybe were not clear, so we have expanded it by: “Now the patient has been HD for ten months, he is in good clinical condition, no signs of infection have been observed and he is prepared for second kidney transplantation”.

Comment 3: 

I would like to thank the authors for acknowledging the study limitation. However, the manuscript presents multiple areas that there is suggestions of what should have done, and whether that would have changed the management course. The latter decreases the scientific soundness of the article. 

Thank you for your comments. We added the suggestion of what should have done in such patients (e.g. minimization of immunosuppression during first period of Covid-19 infection, or treatment with monoclonal antibodies (MAB) against COVID-19) according to the suggestion of 1. Reviewer. In clinical practice not all is clear and we should consider different situations and managements. Perhaps in our case we should have withdrawn from immunosuppressive treatment earlier, but our patient wanted to keep the transplant kidney very much and did not accept a return to dialysis.

Comment 4: 

The patient is CKD by definition having a GFR of 25-35 ml/min/1.73m2. This patient would any way go to dialysis at a point during the coming period. The case doesn't mention whether there was a trend down for the GFR after transplantation, and was there any  drastic change of the GFR drop was subtle. 

Thank you for your comments. The patient had impaired graft function but with stable levels of serum creatinine (2-2.5 mg/dl) and eGFR oscillating 25-35 ml/min/1.73m2 from the beginning after transplantation, for 11 years. We agree that he would any way go to dialysis, at a point during the coming period, but in this case the rapid deterioration of graft function was rather a consequence of a combination of serious infections and their treatment and connected with it complications.
